# Does Focal Osteolysis in a PRECICE Stryde Intramedullary Lengthening Nail Resolve after Explantation?

**DOI:** 10.3390/children9060860

**Published:** 2022-06-09

**Authors:** Oliver C. Sax, Larysa P. Hlukha, Kyle A. Kowalewski, John E. Herzenberg, Philip K. McClure

**Affiliations:** International Center for Limb Lengthening, Rubin Institute for Advanced Orthopedics, Sinai Hospital of Baltimore, Baltimore, MD 21215, USA; osax@lifebridgehealth.org (O.C.S.); kkowalewski@lifebridgehealth.org (K.A.K.); jherzenb@lifebridgehealth.org (J.E.H.); pmcclure@lifebridgehealth.org (P.K.M.)

**Keywords:** osteolysis, limb lengthening, stainless-steel device, intra-medullary lengthening nail, periosteal reaction, Stryde

## Abstract

Concerns surrounding osteolysis near and around the modular junction of a stainless-steel intramedullary lengthening rod prompted a manufacturer recall from the United States market in early 2021. These actions were preceded by similar steps taken in Europe. A concomitant review of stainless-steel lengthenings at our institution demonstrated signs of adverse tissue reaction including periosteal reaction and osteolysis at the modular junction and/or male-sided locking screws. Nearly half of our patients presented with these findings on radiographic images. At the time of the previous review, only half of the nearly 60 implanted stainless-steel devices met a 6-month follow-up. At this juncture, many patients have had their devices explanted. Given the suspected adverse tissue reactions caused by a component of the internal device, we sought to examine the rate of osteolysis post-explantation following removal of a stainless-steel nail. We reviewed a consecutive series of patients who underwent implantation of a stainless-steel limb lengthening device in the femur and/or tibia at a single institution between December 2018 and December 2020. Patients were included if their device was explanted. Periosteal reaction and osteolysis was classified according to a novel and validated classification system, as analyzed by five fellowship-trained surgeons. In addition, changes observed prior to explantation were tracked post-explantation to assess for resolution. The incidence of periosteal reaction and osteolysis prior to explantation was 22/57 (39%) and 15/57 (26%), respectively. Of the 15 patients with osteolysis pre-explantation, 14 patients’ implants were explanted. Of these, eight patients had available follow-up films. Two patients were identified as having partial osteolysis resolution at mean 1-year follow-up, while six patients were identified as having complete osteolysis at mean 18-months follow-up. Periosteal tissue reaction and osteolysis largely resolved following explantation in a subset of patients. These results provide further support to the claim that the stainless-steel device contributed to the changes seen. Further follow-up is warranted to examine the longer-term effects of adverse tissue reaction in this patient population.

## 1. Introduction

Leg length discrepancy (LLD) of less than once centimeter (cm) exists in approximately 90% of the population and can present with a multifactorial complexity of back pain, hip pain, and arthritis when arthritic patients are reviewed retrospectively [1]. However, discrepancies less than 2 cm are rarely treated in a prophylactic manner due to the potential biases in retrospective data analysis, and the fact that the overwhelming majority of patients with minor LLD appear to remain asymptomatic. Most patients with minor LLD have no identifiable cause for their discrepancy. In contrast, congenital femoral deficiency and fibular hemimelia account for a large proportion of patients requiring length-specific corrective surgery [2]. In fact, these conditions can manifest in the ipsilateral limb in up to 68% of cases [3]. Intramedullary limb lengthening devices continue to serve as the preferred management options first brought to the United States (U.S.) in the 1980’s. Ilizarov revolutionized the advancement of correcting LLD utilizing external fixation [4,5]; however, complications such as pin tract infection, joint stiffness, and muscle tethering promulgated the rise of intramedullary devices [6,7,8,9]. These devices have successfully addressed many of the limitations of external fixation, though other inadvertent complications have arisen. A titanium magnetic intramedullary construct (PRECICE, Nuvasive, San Diego, CA, USA) was introduced in 2011, but intra-operative disassembly at the modular junction and limited weight bearing resulted in hesitancy with its adoption [10]. The latest stainless-steel iteration (PRECICE, STRYDE Nail System, Nuvasive) approved in 2018 offered theoretical advantages of early weight bearing and earlier consolidation. However, recent concerns regarding the presence of adverse tissue reaction including periosteal reaction and osteolysis prompted a manufacturer recall in early 2021 [11].

Osteolysis is perhaps most commonly seen and reported in the setting of arthroplasty. Component loosening can be a sequela of macrophage driven periprosthetic osteolysis. Micromotion at the modular junction precipitates wear particles that lead to an inflammatory response with resultant bone resorption. This mechanism is well founded in the arthroplasty literature [12,13,14,15,16,17]. A 2001 report by Jones et al. [18] suggested a similar response with a modular static stainless-steel intramedullary implant in patients who had femoral fractures or non-unions in a trauma setting. The modular junction of stainless-steel intramedullary lengthening nails used for limb lengthening may also be subject to similar changes. In fact, several recent reports have come to light that suggest this process in the stainless-steel intramedullary lengthening device [19,20,21,22,23,24]. However, given these relative newfound insights into the possibility of adverse tissue reactions with this recently introduced lengthening nail, there are no reports describing potential post-explantation resolution.

Despite the reporting of osteolysis limited within hip arthroplasty literature, this process has recently been a subject of concern for the stainless-steel intramedullary lengthening device. As a consequence, the manufacturer recalled this product in early 2021, despite no reports of direct harm to patients [11]. These developments begged the question: if a stainless-steel intramedullary lengthening nail causes adverse tissues reactions, is there resolution after routine explantation? Therefore, at a single tertiary care center, we specifically assessed: (1) incidence of osteolysis before explantation; (2) incidence of osteolysis following explantation; and (3) mean time to osteolysis resolution.

## 2. Methods

### 2.1. Study Design

We reviewed a consecutive series of patients who underwent implantation of a stainless-steel limb lengthening device in the femur and/or tibia at a tertiary care center between December 2018 and December 2020. Local Institutional Review Board (IRB) exemption status was approved for this retrospective review and was in accordance with the International Council for Harmonisation Guidelines for Good Clinical Practice.

### 2.2. Stainless-Steel Intramedullary Lengthening Nail

The stainless-steel intramedullary nail (PRECICE, STRYDE Nail System) was cleared by the Food and Drug Administration in 2018. It is comprised of a magnetic, telescoping high chromium, low nickel stainless-steel alloy (Biodur 108, Pioneer Surgical Technology, Inc., Marquette, MI, USA) nail with solid (male) and cylindrical (female) ends joining at a telescoping junction. A gasket ring enveloping the telescoping junction is a composite of EPDM (ethylene-propylene-diene-monomer) rubber coated with silicone. The system is activated to distract or compress by an external remote controller (ERC). An internal magnet in the rod is induced to rotate by an oscillating external magnetic field, creating distraction through a series of planetary step-down gears linked to a threaded lead screw distraction rod.

### 2.3. Patient Selection

The total number of implanted lower extremities in our study was 57 in 44 patients. In this analysis, patients were included if their device was explanted and if they developed adverse tissue reactions prior to explantation (n = 8). Study exclusion criteria included non-explanted implant, absence of adverse tissue reactions prior to explantation, or implantation with other lengthening devices (titanium lengthening device, extramedullary lengthening device, etc.). Patients with follow-up of less than 12 months, received other treatment methods, or those with serious medical conditions also necessitated exclusion.

### 2.4. Intramedullary Nail Implantation and Explantation Technique

When addressing femoral lengthening, the patient was placed supine and the entire lower extremity was prepped and draped in sterile fashion to include the hip, iliac crest, and gluteal region. Contract scar tissue and fibrous tissue was routinely released from the biceps femoris to the posterior border of the vastus lateralis. This release continued to the iliotibial band and lateral intermuscular septum. Then, a greater trochanter approach was marked. A small stab incision was created at the apex of the deformity on the lateral aspect of the thigh. A 4.8 mm drill bit created multiple drill holes at this level. A Steinmann pin was then inserted at the tip of the greater trochanter towards the center of the proximal femur under fluoroscopy. Reaming over a guidewire was then conducted. Following final reamer, the preoperatively planned nail was inserted and driven past the osteotomy site. The amount of rotation was controlled for using reference Steinmann pins. Once the nail was locked and closing incisions made, we applied the electric remote controller to activate between 0.5 and 2 cm of distraction. Explantation of both femoral and tibial nails were routinely conducted around one-year follow-up from index procedure. Locking screws were retrieved through small stab incisions. We used an ACL reamer over a guidewire to clear the proximal aspect of the nail. Then, an extracting device was inserted and tightened. The nail was removed using a back-slap hammer. Irrigation and sterile closure then followed. 

Tibial lengthening procedures had a similar technique to the above. A tibial and fibular osteotomy were performed through small incision along the posterolateral border of the fibula using a drill bit. Most commonly, a parapatellar starting site was utilized for nail insertion. Blocking screws were typically placed to ensure proper placement of guidewire, reamer, then nail. The nail was then locked proximally and distally, followed by sterile irrigation and closure. Explantation was similar to procedure described above. 

### 2.5. Outcomes of Interest

The primary outcome was presence of adverse tissue reactions after explantation. Adverse tissue reaction was defined as periosteal reaction or osteolysis as classified according to a novel and validated classification system (Biologic Reaction Classification System, as reported by Sax et al. [24]) by five fellowship-trained surgeons. (See Figure 1) This system was developed assessing routine anterior-posterior and lateral standing radiographs for each patient. Secondary outcomes included adverse tissue reactions observed following explantation to assess for resolution. Osteolysis resolution was stratified according to partial and total healing. Partial healing referred to the presence of lytic regions around the intra-medullary device, though these regions were smaller, as compared to pre-explantation radiographs. Total healing referred to the absence of any lystic lesions as compared to prior radiographs, along with consolidation of periosteal reaction. Demographic and baseline characteristic data collection included: patient age, gender, laterality, bone (femur or tibia), clinical follow-up, etiology, and weight. 

Osteolysis resolution was stratified according to partial and total healing. Partial healing refers to the presence of osteolysis, as compared to pre-explantation radiographs, though the lytic regions are lessened. Total healing refers to the absence of any lytic lesions as seen on prior radiographs.

### 2.6. Statistical Analyses

All variables including baseline demographics, indications for procedures, and adverse tissue reactions were descriptively tallied and listed as a percentage of all implanted stainless-steel intramedullary lengthening nails. Table formation was completed using Microsoft Excel (Seattle, WA, USA). 

## 3. Results

### 3.1. Demographics and Baseline Characteristics

Of all stainless-steel implants (n = 57), the mean age was 15.6 years (range 10–18 years), mean weight was 63 (range, 48 to 82 kg), with male predominance (43/57). (See Table 1). Most were implanted on the right side (37/57) and the most common bone treated was the femur (38/57). Most had LLD due to congenital diseases (48/57).

### 3.2. Adverse Tissue Reactions Prior to Explantation

The incidence of periosteal reaction and osteolysis prior to explantation was 22/57 (39%) and 15/57 (26%), respectively.

### 3.3. Adverse Tissue Reactions Post-Explantation

Of the 15 patients with osteolysis pre-explantation, 14 patients’ implants were explanted. Of these, eight patients had available follow-up films (See Table 2 and Table 3). Two patients were identified as having partial osteolysis resolution at mean 1-year follow-up, while six patients were identified as having complete osteolysis at mean 18-months follow-up. Of note, none of the patients who were identified with adverse tissue reactions developed any inadvertent medical or surgical complications, including thigh or lower leg pain, early removal of hardware, mechanical failure, or infection. 

## 4. Discussion

The modular components within the stainless-steel intramedullary limb lengthening nail are subject to produce wear particles and a macrophage-driven adverse tissue reaction. Reports of osteolysis between similar designs are largely limited to total hip arthroplasty (THA); though recent reports have suggested this process in intramedullary lengthening nails [19,20,21,22,23,24]. We sought to examine the total number of stainless-steel nail recipients for LLD at a single tertiary care center to determine the incidence of adverse tissue reaction before and following routine explantation. Periosteal reaction and osteolysis developed in 39% and 26% in our studied population, respectively. At the time of analysis, eight patients have had their implants explanted and they all demonstrated either partial or complete resolution of their adverse tissue reactions. Patients number 3 and 8 in Table 3 showed partial resolution at 6 and 7 months, respectively. The rest of our assessed cohort exhibited full resolution of osteolysis at 9 months. Since these are the most recent radiographs, we expect full resolution at a later follow-up date.

This study has several limitations. The primary outcome of adverse tissue reaction was defined according to the Biologic Reaction Classification System, the first system to identify the progression of periosteal reaction and osteolysis in intramedullary nails with a modular junction. Despite early validation of this system by five fellowship trained orthopaedic surgeons, it was only recently reported and has not been applied to other datasets. Another limitation is the sample size. Our institution implanted 57 stainless-steel nails and only eight recipients had reported adverse tissue reactions prior to explantation and were explanted at the current iteration of this study. Longer follow-up will account for the remaining individuals who were identified with adverse tissue reactions post-explantation. These limitations should not disqualify the strength of this study, namely that this is the first attempt to identify the progression and resolution of adverse tissue reactions in post-explanted recipients. 

The art of limb lengthening was popularized in the 1980’s by Ilizarov with the use of external fixation assisted devices in multiple phases [4,5]. In the first phase, the bone would be progressively distracted to achieve length desired. In the second phase, the distracted bone completed consolidation, thereby signaling a healing milestone for the new bone. However, Ilizarov’s initial reports of patients treated detail a high proportion of unplanned procedures during the lengthening process with some patients stopping the lengthening due to other concerns such as pin tract infection, stiffness, and muscle tethering. However, many of the complications reported using the Ilizarov techniques are specific to the use of external fixation. The rise of intramedullary limb lengthening nails nearly eliminated these complications but using similar techniques. Several studies over the past few decades have assessed the techniques promulgated by Ilizarov as compared to management using intramedullary lengthening nails. In a matched comparison of 19 tibial lengthenings using intramedullary nails versus 19 Ilizarov lengthenings, Burghardt et al. [25] demonstrated slightly shorter time to consolidation (6.6 versus 7.6 months) among the intramedullary cohort, but increased blood loss and costs. In a more recently introduced technique, Rozbruch et al. [6] assessed the use of lengthening and then nailing (LATN), whereby the shortened limb is distracted using a frame, following by reaming and insertion of an intramedullary nail to support the bone during consolidation. When comparing this cohort to classically treated patients (external fixation only), they observed shorter external fixation wear time (12 versus 29 weeks), and a lower bone healing index (0.8 versus 1.9) among the LATN cohort. Though these studies demonstrate preservation of treatment successes with relative elimination of external fixation related complications, inadvertent consequences undoubtedly arise. The earlier intramedullary lengthening devices were associated with mechanical failures, including intra-operative disassembly, failure to activate the device, and breakage of the device [10,26]. A more recent iteration of a lengthening device made of stainless steel has drawn recent negative attention given its reports of adverse tissue reaction surround its modular junction.

Osteolysis is a well-known phenomenon following the production of metal particulate debris from modular components, but reporting has been largely limited to arthroplasty literature [12,13,14,15,16,17]. The mechanism and manifestations of osteolysis around an intramedullary device were first well-publicized in the 1990’s and early 2000’s. In a careful pathologic review of 34 hips with a prosthetic replacement, Schmalzried et al. [27] assessed patterns of bone loss. They found extracellular and intracellular particulate debris of polyethylene in 34 (100%) and 31 hips (91%), respectively. They also found that the number of macrophages observed were directly related to the bone resorption seen on radiographs. Jacobs et al. [28] characterized the osteolytic process by reviewing the plethora of chemical mediators responsible. Earlier reports of osteolysis in the setting of hip arthroplasty describe a polyethylene particular process, though the influence of stainless-steel metal has also been suggested to contribute to metal debris-associated osteolysis. In a review of 27 mid-diaphyseal femoral fractures treated with a modular stainless-steel femoral intramedullary nail, Jones et al. [18] observed that 23 fractures had at least one of the three reactions: osteolysis, periosteal reaction, or cortical thickening. Retrieval analysis demonstrated signs of fretting corrosion, stainless-steel corrosion products adherent to the modular junction, and a foreign-body granulomatous result. In addition, these findings were associated with increased serum chromium level. Importantly, a comparison group of non-modular stainless-steel implants did not demonstrate the corrosion or chromium levels, drawing attention to the modular component as the culprit of the observed adverse tissue reactions. 

Since a manufacturer recall in early 2021, multiple reports have demonstrated the presence of adverse tissue reactions in stainless-steel intramedullary lengthening nails [19,20,21,22,23,24,29,30]. In a report of three cases, Sax et al. [19] demonstrated radiographic osteolysis in patients implanted with the stainless-steel nail with histologic confirmation of phagocytic activity of green-gold crystalline debris characteristic of metal debris. Other institutional studies have also found adverse tissue reactions in a substantial proportion of their patients. In a review of 13 implants between June 2019 and September 2020 by Iliadis et al. [21], osteolysis and periosteal reaction at the modular junction were observed in nine implants. Focal metallic wear debris was also noted on histological analysis. A similar report by Frommer et al. [20] identified osteolytic changes in 20 of the 57 (35%) implanted nails at a mean 9.5 months after index lengthening surgery. They also found macroscopic corrosion at the modular junction in 83% of recipients, similar to that reported by our group in a previous publication [19] and a separate analysis of retrieved nails at another institution [30]. In a comparative study of multiple variations of an intramedullary limb lengthening nail, Iobst et al. [29] observed a greater proportion of bone abnormalities at the modular junction of the STRYDE nail, as compared to the PRECICE or FITBONE (Orthofix Medical Inc., Lewisville, TX, USA). Given the current clinical understanding and relative dearth of data surrounding osteolysis at the modular junction of intramedullary limb lengthening nails, the existing reports are limited to descriptions of a pre-explantation process. The present study findings corroborate the results reported by earlier authors and extends the analysis of adverse tissue reactions to the post-explantation phase. Of all eight patients assessed, partial or complete resolution of the presence of adverse tissue reaction occurred. These results support the speculation that the reactions were sequelae of metal debris precipitated by the modular junction of the lengthening nail. However, this is the first report to detail the preliminary resolution of osteolysis in post-explanted limb lengthening nails. Of note, the adverse tissue reactions reported in this series and by other authors are not associated with concerning patient complications such as severe thigh pain, infection, or mechanical failures. Though this is reason for much optimism, we encourage close follow-up on patients implanted with the stainless steel intramedullary limb lengthening device to evaluate for subtle clinical or radiographic changes that may arise. 

## 5. Conclusions

A substantial proportion of our patients who underwent limb lengthening with a stainless-steel nail developed adverse tissue reactions. However, of the eight patients who developed these findings and were explanted, all had either partial or complete resolution at three months follow-up. The clinical implications of these findings suggest that orthopaedic surgeons and families alike should be reassured that if adverse tissue reactions develop during the lengthening process, they are likely to be resolving or resolved in the post-explantation period. Longer follow-up will be needed to confirm these observations. 

## Figures and Tables

**Figure 1 children-09-00860-f001:**
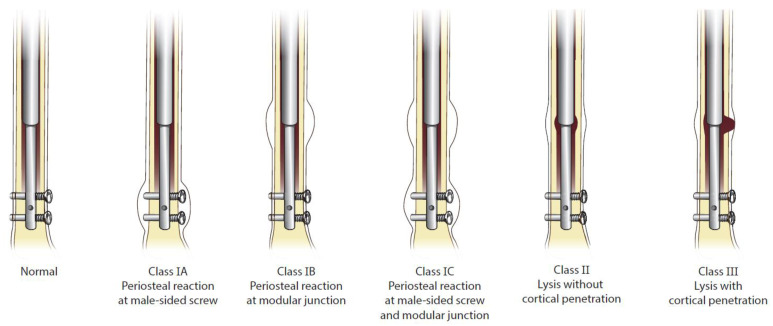
Biologic Reaction Classification System.

**Table 1 children-09-00860-t001:** Demographics.

Mean Age	15.6
Gender (M, F)	43 M, 14 F
Laterality (L, R)	37 R, 20 L
Bone (Femur, Tibia)	38 Femurs, 19 Tibias
Average Clinical Follow-up Time (Months)	5.6
Etiology	48 Congenital, 5 Idiopathic, 4 Traumatic
Average Weight (kg)	63

**Table 2 children-09-00860-t002:** Adverse Tissue Reaction (ATR) Pre-Explantation.

Patient *	Osteolysis Onset (mo)	Time to Explantation (mo)	Periosteal Reaction Class Prior to Explanation (1A/1B/1C)	Osteolysis Class Prior to Explanation (II/III)
1	13	18	1A	II
2	10	14	1C	II
3	20	22	1C	II
4	9	33	1C	II
5	6	Not explanted/lost to f/u	1A	II
6	12	20	1A	II
7	9	11	1A	II
8	2	5	1C	III
9	14	19	1C	II
10	8	16	1C	II
11	10	23	1A	II
12	23	24	1B	II
13	5	11	1C	II
14	6	7	1B	II
15	6	7	1B	II

f/u: follow-up. * 15 total patients with pre-explantation ATR.

**Table 3 children-09-00860-t003:** Post-Explantation Osteolysis Resolution.

Patient *	Last Clinical f/u (mo)	Last Clinical f/u Osteolysis Classification (II/III)	Osteolysis Resolution
1	9	II	Complete
2	9	II	Complete
3	6	II	Partial
4	13	II	Complete
5	No f/u	II	n/a
6	14	II	Complete
7	not explanted		
8	7	III	Partial
9	No f/u	II	n/a
10	No f/u	II	n/a
11	No f/u	II	n/a
12	No f/u	II	n/a
13	No f/u	II	n/a
14	10	II	Complete
15	12	II	Complete

f/u: follow-up. n/a: not applicable. * of the 15 patients, 14 were explanted.

## Data Availability

The data presented in this study are available upon request from the corresponding author. The data are not publicly available due to privacy concerns with protected health information.

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
