# Peer review of "Does Focal Osteolysis in a PRECICE Stryde Intramedullary Lengthening Nail Resolve after Explantation?"

_children, 2022, doi:10.3390/children9060860_

Round 1

Reviewer 1 Report

This study is a retrospective series of bone lengthening cases using the STRYDE IM nail investigating bone reaction. It bears several limitations, but the topic is important. Therefore, it should be published after revisions.  

-Title: The study involves the STRYDE nail, not any SS lengthening nail. It should appear namely.

-Introduction. page 2 line 55. ‘no reports of direct harm to patients’. The paper cited as Ref 18 reports on 6 patients with late onset pain, of which 2 were unable to fully bear weight despite healing of the regenerate. I would consider this harmful if this term is to be used.

-Material and methods. Page 2 line 62. Again, the device investigated must appear namely here.

-Material and methods. Page 2 line 78. Define age limit for ‘pediatric patients’  

-Material and methods. If the inclusion criteria listed are applied, the study group should only count 8 patients, not 15 (1 not explanted, 6 with no FU post explantation). Revise accordingly.

-Results. Page 3 line 97. Provide age range and weight range.

-Conclusion. Page 5 line 175. This should read 8 patients, not 5.

-Conclusion. Page 5 line 178. The results of the current study are not robust enough to reassure either surgeons or patients yet. They only suggest that bone reaction is reversible after implant removal.

Author Response

Reviewer 1 Response.

Comments and Suggestions for Authors

This study is a retrospective series of bone lengthening cases using the STRYDE IM nail investigating bone reaction. It bears several limitations, but the topic is important. Therefore, it should be published after revisions.  

Dear Reviewer 1, thank you for your comments.

-Title: The study involves the STRYDE nail, not any SS lengthening nail. It should appear namely.

Response: The title has been updated with a manufacture name of the nail

-Introduction. page 2 line 55. ‘no reports of direct harm to patients. The paper cited as Ref 18 reports on 6 patients with late onset pain, of which 2 were unable to fully bear weight despite healing of the regenerate. I would consider this harmful if this term is to be used.

Response: We agree with the statement. Although reports of concurrent osteolysis, periosteal reaction and other adverse events such as delayed post-operative pain and delayed healing exist, literature does not directly link these events. Reference 18 mentions that delayed onset of pain could have been the prodrome to developing osteolysis. The statement in line 55 was removed for clarification.

-Material and methods. Page 2 line 62. Again, the device investigated must appear namely here.

Response: The changes to device name were made in lines 67-68

-Material and methods. Page 2 line 78. Define age limit for ‘pediatric patients’  

Response: The reference ages were defined and mean calculated on line 81. This was updated in Table 1: Demographics

-Material and methods. If the inclusion criteria listed are applied, the study group should only count 8 patients, not 15 (1 not explanted, 6 with no FU post explantation). Revise accordingly.

Response: After accounting for inclusion criteria, 8 total patients were analysed. This is reflected in line 82 of the Methods section. In the result section (lines 117-119), it is documented that 15 patients had pre-explantation osteolysis but only 8 had sufficient follow-up films.  

-Results. Page 3 line 97. Provide age range and weight range.

Response: Age and weight range provided in results lines 109-110. The means have also been updated in Table 1: Demographics.

-Conclusion. Page 5 line 175. This should read 8 patients, not 5.

Response: This has been edited and now appears in line 166. Thank you

-Conclusion. Page 5 line 178. The results of the current study are not robust enough to reassure either surgeons or patients yet. They only suggest that bone reaction is reversible after implant removal.

Response: Thank you for the comments. We have modified wording of the conclusion to reflect that larger studies with longer follow-ups are needed to confirm compete resolution of osteolysis post-explanation and provide re-assurance.

Reviewer 2 Report

In general, the manuscript tackles a trending theme of great importance. Is a short and objective retrospective study (n=8) set on radiography imaging FU. However, some sections should be improved.

Introduction:

Short, objective and well written.

I suggest putting the dots right after the citation (e.g. [1,2].) across the entire paper.

A small paragraph contextualizing the limb-length discrepancy, the surgical procedure and pediatrics would make sense (e.g. age of diagnostics and surgery, common etiology, incidence, correction’s benefits), once the journal aims relevant topics to children’s health.

Methods/results:

Line 81 – a dot is missing

Authors described the system and specified that is made with Biodur stainless steel. However, the name of the intramedullary nail system analyzed in the study was not explicitly identified by the authors. PRECICE, STRYDE Nail System (Nuvasive) was mentioned in the introduction. Was any other stainless steel based system included in the study? I suggest pointing out the name of the system in the methods section.

I have some doubts regarding the differences of the primary and the secondary outcomes – “primary outcome was presence of adverse tissue reactions after explantation” // “Secondary outcomes included adverse tissue reactions observed following explantation to assess for resolution.” I suggest rewriting the sentence or give examples of the secondary outcomes to ease the understanding. Furthermore, the presence of adverse tissue reactions before explantation as an outcome would be useful and comparable to the literature.

Was the systemic health of the patients taken into consideration in the present study? Even though most patients were young, the usage of medications and/or systemic diseases can play an important role in chronic inflammation and, therefore, osteolysis. I suggest the addition of this information in table 1 or in the inclusion/exclusion criteria.

Describe partial and total healing of osteolysis parameters.

Tables:

Table 2 is written “explanation” in instead of explantation. The subtitle mentions post explantation in table 2, however, only pre explantation data is put.

Patient n° 7 is classified as class II in table 2 and as class III in table 3. Is it correct or the osteolysis got worse after the removal surgery/last FU? Patient n° 3 has this same difference (classified as III in table 2 and II in table 3).

I suggest merging tables 2 and 3 – easier to compare pre-explantation parameters with the outcome.

Line 98 – a dot is missing, after (table 1).

Discussion/conclusion:

Line 125 – insert reference

Partial resolution patients have the last FU of 6 and 7 months, while the complete have, at minimum, 9 months. In my opinion, authors should address that in the discussion.

Line 162/163 - insert reference

Line 169 and line 175 – “Of all five patients assessed…” - Was the final number of assessed patients five or eight? For my reading, 5 patients complained about the implant prior the explantation. Nevertheless, 8 patients had the explant surgery.

In the conclusion is written: “…all had either partial or complete resolution at three months follow up.” However, I could not find this information in the results nor in the discussion section. No new information should be inserted in the conclusion. In addition, this information do not match the abstract’s information. 

Author Response

 Reviewer 2 Response

Comments and Suggestions for Authors

In general, the manuscript tackles a trending theme of great importance. Is a short and objective retrospective study (n=8) set on radiography imaging FU. However, some sections should be improved.

Dear Reviewer 2, thank you for your comments.

Introduction:

Short, objective and well written.

I suggest putting the dots right after the citation (e.g. [1,2].) across the entire paper.

Response: This edition has been implemented throughout the paper.

A small paragraph contextualizing the limb-length discrepancy, the surgical procedure and pediatrics would make sense (e.g. age of diagnostics and surgery, common etiology, incidence, correction’s benefits), once the journal aims relevant topics to children’s health.

Response: Thank you for the helpful comment. A brief paragraph was added in the introduction regarding incidents and treatment options for limb length discrepancy in children.

Methods/results:

Line 81 – a dot is missing

Response: thank you, an edition has been made.

Authors described the system and specified that is made with Biodur stainless steel. However, the name of the intramedullary nail system analyzed in the study was not explicitly identified by the authors. PRECICE, STRYDE Nail System (Nuvasive) was mentioned in the introduction. Was any other stainless steel based system included in the study? I suggest pointing out the name of the system in the methods section.

Response: Thank you for the comment. This study assessed osteolysis and periosteal reaction with Stryde stainless steel nails only. Only patients who undergone lengthening with a Stryde nail were included. This was further clarified in the methods, introduction and results of the manuscript.  (lines 57, 59, 68, 73)

I have some doubts regarding the differences of the primary and the secondary outcomes – “primary outcome was presence of adverse tissue reactions after explantation” // “Secondary outcomes included adverse tissue reactions observed following explantation to assess for resolution.” I suggest rewriting the sentence or give examples of the secondary outcomes to ease the understanding. Furthermore, the presence of adverse tissue reactions before explantation as an outcome would be useful and comparable to the literature.

Response: The primary outcome of this study was to assess adverse tissue reactions (i.e., periosteal reaction, osteolysis) after explantation of the intramedullary limb lengthening nail. Secondary outcomes included timing of post-explantation adverse tissue reaction resolution, as well as the presence of adverse tissue reactions prior to explantation. The edition was made on line 96 in the methods section and highlighted in Table 2 as periosteal reaction prior to explantation.

Was the systemic health of the patients taken into consideration in the present study? Even though most patients were young, the usage of medications and/or systemic diseases can play an important role in chronic inflammation and, therefore, osteolysis. I suggest the addition of this information in table 1 or in the inclusion/exclusion criteria.

Response: Thank you for your comment. Patients possessing serious comorbidity or requiring life saving medication were not included in this study. The materials and methods regarding this were updated (lines 88-89).

Describe partial and total healing of osteolysis parameters.

Response: Osteolysis resolution was stratified according to partial and total healing. Partial healing refers to the presence of osteolysis, as compared to pre-explantation radiographs, though the lytic regions are lessened. Total healing refers to the absence of any lytic lesions as seen on prior radiographs. This clarification was added into lines 98-100.

Tables:

Table 2 is written “explanation” in instead of explantation. The subtitle mentions post explantation in table 2, however, only pre explantation data is put.

Response: Thank you for the comment, we have corrected it.

Patient n° 7 is classified as class II in table 2 and as class III in table 3. Is it correct or the osteolysis got worse after the removal surgery/last FU? Patient n° 3 has this same difference (classified as III in table 2 and II in table 3).

Response: Table 3 originally only reflected those patients who were explanted. Since 1/15 was not explanted, Table 3 only had 14 patients total, hence changing the numbers from that of Table 2. We have modified Table 3 to reflect the explanted patient and provide corresponding patient numbers between Table 2 and 3.

Patient #3 shows osteolysis class II in both tables

I suggest merging tables 2 and 3 – easier to compare pre-explantation parameters with the outcome.

Response: Thank you for your comment. We attempted merging the two tables, however, it is important to differentiate pre-and post-explantation reaction to monitor resolution. The merged tables, to author’s opinion are too busy for sufficient comparison.

Line 98 – a dot is missing, after (table 1).

Response: This was edited (now line 110)

Discussion/conclusion:

Line 125 – insert reference

Response: The first paragraph of discussion is summarizing the results we achieved in our studied cohort. Therefore, there is no reference needed to these percentages.

Partial resolution patients have the last FU of 6 and 7 months, while the complete have, at minimum, 9 months. In my opinion, authors should address that in the discussion.

Response: Thank you for the comment. This is an astute observation and should be further elaborated in the discussion section as per your suggestion. A paragraph discussing this was added in lines 133-136.

Line 162/163 - insert reference

Response: References has been inserted (now lines 170)

Line 169 and line 175 – “Of all five patients assessed…” - Was the final number of assessed patients five or eight? For my reading, 5 patients complained about the implant prior the explantation. Nevertheless, 8 patients had the explant surgery.

Response: In total there were eight patients with sufficient follow-up post explantation to assess for osteolysis/periosteal reaction resolution. Indication for explantation was not presence or absence of pain but rather a nationwide recall of the Stryde nail due to its propensity to cause adverse reactions. Thank you for noticing this discrepancy. The numbers have now been updated.

In the conclusion is written: “…all had either partial or complete resolution at three months follow up.” However, I could not find this information in the results nor in the discussion section. No new information should be inserted in the conclusion. In addition, this information do not match the abstract’s information. 

Response: Thank you for providing this comment. We have indeed noticed osteolysis beginning to resolve at 3 months follow up. However, some patients did not have follow-up radiographs at those time marks. Therefore, we decided to standardize the results and claim that at least partial resolution was seen in all patients at 6 months follow-up. Since this device is now off the market, and many patients don’t exhibit any symptoms of adverse tissue reaction, it is a challenge to standardize regular radiographic follow-up. Ideally, we would like to have a larger volume of patients to compare these results to.

Reviewer 3 Report

Thank you for submitting this paper to Children. The manuscript under consideration: "Does Focal Osteolysis in a Stainless-Steel Lengthening Device Resolve after Explantation?" is an interesting article on an important topic in Children. However, there are a few minor concerns.

1. How did the authors determine the sample appropriate size? Why not power calculations performed a priori? Please provide all parameters for the sample size calculation in the Methods.

2. How was the data stored ? Consider a flow diagram regarding recruitment process and data collection.

3. Why didn't you consider the effect of gender in this study?

Author Response

  1. How did the authors determine the sample appropriate size? Why not power calculations performed a priori? Please provide all parameters for the sample size calculation in the Methods.

Dear Reviewer 3, thank you for your comments.

Response: No power analysis was conducted for this study. Given that the adverse tissue reaction concern has only been introduced within the past year, a proper sample size analysis may not be possible.

  1. How was the data stored? Consider a flow diagram regarding recruitment process and data collection.

Response: The 57 patients analyzed in this study have all received a Precice Stryde nail to treat limb length discrepancy. All the nails were inserted by surgeons from a single institution who are well-versed in limb lengthening procedures. Data then was collected retrospectively utilizing patient’s clinical chart and stored and analyzed using Excel. This information is reflected in the Method section under Statistical Analysis.

  1. Why didn't you consider the effect of gender in this study?

Response: This is a very interesting point, since we have three times more Males who received these nails vs. Females, it could be of interest to stratify patients based on gender for subsequent studies. Throughout existing literature, we did not find correlation with gender and osteolysis development, however, it is something to account for.